# Prognostic Value and Immune Infiltration of HPV-Related Genes in the Immune Microenvironment of Cervical Squamous Cell Carcinoma and Endocervical Adenocarcinoma

**DOI:** 10.3390/cancers15051419

**Published:** 2023-02-23

**Authors:** Qiyu Gan, Luning Mao, Rui Shi, Linlin Chang, Guozeng Wang, Jingxin Cheng, Rui Chen

**Affiliations:** 1Department of Gynecology and Obstetrics, Shanghai East Hospital, School of Medicine, Tongji University, Shanghai 200120, China; 2Key Laboratory of Carcinogenesis and Translational Research (Ministry of Education/Beijing), Department of Pathology, Peking University Cancer Hospital & Institute, Beijing 100021, China; 3Department of Gynecology, Shanghai United Family Hospital, Shanghai 200120, China

**Keywords:** cervical squamous cell carcinoma and endocervical adenocarcinoma, TCGA datasets, immune microenvironment, HPV

## Abstract

**Simple Summary:**

Cervical squamous cell carcinoma and endocervical adenocarcinoma (CESC) generally presents with HPV infection and is the second most common gynecological malignancy. However, the effect of immune infiltrate and immune microenvironment on the tumorigenesis and development of CESC remains unclear. In this study, we divided CESC cases into different immune subtypes and performed a differential gene expression analysis. The CESC cases (*n* = 303) were divided into five subtypes (C1–C5) based on their expression profiles. Subtype C4 demonstrated a downregulation of the immune profile, lower tumor immune/stroma scores, and worse prognosis, while subtype C1 showed the opposite characteristics. In addition, GSEA screened out some key genes associated with HPV infection pathways, among which high FOXO3 and low IGF-1 protein expression were closely correlated with decreased clinical prognosis. Our results may provide guidance for developing potential immunotherapeutic targets and biomarkers for CESC.

**Abstract:**

Mounting evidence has highlighted the immune environment as a critical feature in the development of cervical squamous cell carcinoma and endocervical adenocarcinoma (CESC). However, the relationship between the clinical characteristics of the immune environment and CESC remain unclear. Therefore, the aim of this study was to further characterize the relationship between the tumor and immune microenvironment and the clinical features of CESC using a variety of bioinformatic methods. Expression profiles (303 CESCs and three control samples) and relevant clinical data were obtained from The Cancer Genome Atlas. We divided CESC cases into different subtypes and performed a differential gene expression analysis. In addition, gene ontology (GO) and gene set enrichment analysis (GSEA) were performed to identify potential molecular mechanisms. Furthermore, data from 115 CESC patients from East Hospital were used to help identify the relationship between the protein expressions of key genes and disease-free survival using tissue microarray technology. Cases of CESC (*n* = 303) were divided into five subtypes (C1–C5) based on their expression profiles. A total of 69 cross-validated differentially expressed immune-related genes were identified. Subtype C4 demonstrated a downregulation of the immune profile, lower tumor immune/stroma scores, and worse prognosis. In contrast, the C1 subtype showed an upregulation of the immune profile, higher tumor immune/stroma scores, and better prognosis. A GO analysis suggested that changes in CESC were primarily enriched nuclear division, chromatin binding, and condensed chromosomes. In addition, GSEA demonstrated that cellular senescence, the p53 signaling pathway, and viral carcinogenesis are critical features of CESC. Moreover, high FOXO3 and low IGF-1 protein expression were closely correlated with decreased clinical prognosis. In summary, our findings provide novel insight into the relationship between the immune microenvironment and CESC. As such, our results may provide guidance for developing potential immunotherapeutic targets and biomarkers for CESC.

## 1. Introduction

Cervical squamous cell carcinoma and endocervical adenocarcinoma is a malignant tumor occurring in cervical epithelial cells, with more than 500,000 women diagnosed per year [1]. In addition, over 300,000 patients per year die as a result of the disease [1,2]. Among the majority of CESC cases, high-risk subtypes of the human papilloma virus (HPV) are considered to be the primary cause [1,3]. Despite a standardized treatment involving radical hysterectomy, chemoradiation, or a combination [4,5], the mortality and recurrence rates of CESC continue to increase [4]. Recent studies have demonstrated that tumor heterogeneity and immune microenvironment [6,7] act in carcinogenesis and the epithelial–mesenchymal transition (EMT). However, the molecular mechanisms underlying the role of the immune microenvironment in CESC remain unclear.

CESC is primarily considered to be a typical immunogenic cancer as a result of HPV infection, and much attention has been paid to immunotherapy treatments for this disease. As such, monoclonal antibodies (mAb) that target various immune checkpoint pathways have been approved for the treatment of CESC by the U.S. Food and Drug Administration (US FDA) [8]. Indeed, CESC is characterized by several alterations in the immune system, indicating the potential for patients to benefit from immunotherapy. Recently, the U.S. FDA granted approval for the use of pembrolizumab in patients who experienced disease progression during or after chemotherapy in cases of programmed death-ligand 1 (PD-L1) positive expression [9]. However, the benefit of immunotherapy in primary CESC remains unknown.

Previous studies have shown that various features of the tumor microenvironment in CESC are crucial in the immune process of carcinogenesis and progression. To illustrate the underlying immune gene expression profiles, the prognosis of CESC patients has been previously observed by quantitative PCR (qPCR) and immunohistochemistry (IHC) [10]. A gene expression profile analysis of 27 immune response genes and 15 vascularized marker genes in 56 CESC samples has been performed by weighted gene coexpression network analysis. However, further studies with a larger sample size would have increased the reliability of the conclusions. In another study, 27 survival-related immune response genes and 15 vascularized marker genes were elucidated from six tumor clusters (four immune response clusters and two vascularization clusters) in 52 CESC samples using qPCR and IHC techniques [11]. With these types of high-throughput approaches, comprehensive analysis of the tumor molecular types and global immune profiles are necessary. The transcriptome serves as a powerful tool for systematically exploring the tumor microenvironment, which is infiltrated with immune cells such as T cells, B cells, and dendritic cells (DCs), and involved in tumor initiation, growth, angiogenesis, metastasis, and tumor immunity [12,13]. In addition, several supporting studies have suggested RNA sequencing and gene expression data from underlying expression profiles may explain the correlations between heterogeneous clinical outcomes and coexistent tumor immune microenvironments. Using microarray analysis, qPCR, and Western blot techniques, an immune-related signature of the microenvironment has been identified as a potential therapeutic target of early-stage HPV infection in the cervical cancer tissue of CESC [14]. However, owing to the lack of external validation and cross-validation in the training set of this previous study, studies with a larger number of patients with CESC are required for the findings to be further validated.

In this study, we present the analysis of patients with CESC from a cohort available from The Cancer Genome Atlas (TCGA) in order to explore relevant immune subtypes and their clinical associations with the immune microenvironment. RNA sequencing data were used to investigate the correlation between the immune score of CESC subtypes and clinical information. In addition, we analyzed the immune infiltration level of key genes that were differentially expressed in the immune microenvironment. Furthermore, the correlations between the expression of key genes and tumor progression/regression status was validated by tissue specimens from 115 CESC patients using tissue microarray technology. The results of our study should provide additional insight into the mechanism of immunotherapy in CESC.

## 2. Materials and Methods

### 2.1. Data Acquisition

RNA-seq data of CESC were retrospectively retrieved from the TCGA (https://tcga-data.nci.nih.gov/tcga/ accessed on 9 January 2022) [15]. Corresponding clinical information, including age, pathological type, survival, and prognosis were also obtained from TCGA database and integrated with RNA-seq data. The criteria were as follows: (1) stage I, II, III, or IV; (2) with complete survival data; and (3) accompanied by the gene expression pattern. Gene expression data were preprocessed from limma (R package) using quantile normalization [16]. Stromal and immune scores were calculated using ESTIMATE algorithm for tumor purity [17]. No research was performed on patients, participants, or animals by any of the authors in this part of the current study.

There was a total of 115 Chinese patients diagnosed with CESC from January 2010 to December 2011 in Shanghai East Hospital who were included in this study. The average age was 46.574 (range, from 29 to 70 years). The inclusive criteria were (1) a pathologic diagnosis of squamous cell cancer; (2) complete surgical section of the primary tumor and regional lymph node, with negative margins by histologic examination; (3) entire clinicopathologic testing; and (4) follow-up information. Exclusion criteria included the administration of neoadjuvant chemotherapy before surgery. All patients had signed the written informed consent forms before surgical resection. This clinical part of our study was approved by the Ethics Committee of the Shanghai East Hospital, which is affiliated with the Tongji University of Shanghai. Table 1 lists the detailed clinicopathologic features of the CESC patients.

### 2.2. Data Processing

RNA-seq data were obtained using an Illumina^®^ system, which is a next-generation sequencing platform [18]. We downloaded the fragments per kilobase of gene per million fragments mapped with upper quartile normalization (FPKM-uq) from the TCGA database, converted the gene annotation using the Ensemble database [19], and transformed the gene expression value into log^2^ for further analysis. The ESTIMATE algorithm was used to calculate tumor purity and the total immune component in each CESC sample using the downloaded gene expression data [20]. The scores of six immune infiltration subtypes were downloaded from the Timer database (https://cistrome.shinyapps.io/timer/ accessed on 1 March 2022), including those of B cells, T cells (CD8^+^ T cells, CD4^+^ T cells), macrophages, neutrophils, and DCs [21].

### 2.3. Immunogene-Based CESC Molecular Subtypes

Using the ConsensusClusterPlus R package [22], we performed consensus clustering to identify the molecular subtypes of CESC underlying the immune gene expression profiles. Then, we selected immune-related genes from the Gene Ontology database [23] by searching immune-related GO terms. Gene expression data were median centered. Using the euclidean distance, we analyzed the similarity distance between all samples [24]. A clustering program based on K-means [25] was then carried out with 1000 iterations by extracting 80% of the samples at each iteration. Through CDF curves of the consensus score, the optimal cluster number was achieved [26]. In addition, SigClust analysis in pairwise comparisons was employed to illustrate the significance of clustering among the classified subtypes [27]. Bonferroni correction [28] was applied for multiple comparison testing. Characteristic genes were selected by the sample comparison in each subtype, and Student’s *t*-test was used for the remaining samples. Different molecular subtypes within each subtype were distinguished. The Benjamini–Hochberg procedure was applied to calculate the false discovery rate (FDR) [29].

### 2.4. Differentially Expressed Genes

A differential gene expression analysis was carried out using limma (R package) [30]. We corrected the DEGs using FDR adjustment for multiple comparisons. Limma [31] was also used to screen for DEGs based on each subtype. The cut-off criteria were set to fold change >−1 with an adjusted *p*-value <0.05. The results for the top 100 upregulated genes were used to distinguish the different molecular subtypes. Subsequently, the top 100 upregulated DEGs from each subgroup were intersected using a Venn diagram, and overlapped DEGs were obtained.

### 2.5. Functional Enrichment

GO and Kyoto encyclopedia of genes and genomes (KEGG) gene expression profiles of the DEGs were used to identify GO categories [32] and relevant pathways of protein networks, chemical information, and genomic information [33]. The GO term and KEGG pathway analyses were carried out using the clusterProfiler method (R package) [34]. An FDR < 0.05 was regarded to be significant.

### 2.6. Gene Set Enrichment Analysis

The Reactome pathways of the DEGs were explored by GSEA [35] using clusterProfiler (R package). The reference gene set of C2.all.v6.2.symbols.gmt was investigated in the study. The cut-off criteria were set as FDR < 0.1 and a *p*-value <0.01.

### 2.7. Survival Analysis

The mean survival time of the samples was estimated by using the Kaplan–Meier method. Survival curves were assessed using the log-rank test. All tests were two-sided and were executed using the R Version 3.6.1 statistical software [36]. The cut-off criteria were set as *p*-value <0.05.

### 2.8. Immunohistochemistry

Using tissue microarray, all samples were fixed in 4% paraformaldehyde at 4 °C overnight. Five micrometer-thick histological sections were processed using ethanol dehydration, xylene clearing, and paraffin embedding. Sections were incubated with primary antibodies (anti-*PERP*, -*BAK1*, -*CDK2*, -*VDAC1*, -*MDM2*, -*DAC1*, -*FOXO3*, -*AKT3*, and -*IGF1*; 1:100; Bioss, Beijing, China) at 4 °C overnight. The staining procedure was performed according to the instruction of the commercial kit (ZsBio, Beijing, China).

The IHC analysis was performed by two independent pathology investigators at 400× magnification in five randomly selected representative fields, separately. A semiquantitative scoring system was applied to the assessment. The criteria were as follows: (1) staining intensity: 0, no staining; 1+, weak staining; 2+, moderate staining; 3+, strong staining; (2) percentage of stained cells: 0, <5%; 1, 5–25%; 2, 26–50%; 3, 51–75%; and 4, >75% [37].

### 2.9. Statistical Analysis

Samples were stratified into high or low expression by the threshold of median value. Data were presented as the mean ± standard deviation. The possible differences of the key genes were analyzed using an independent samples *t*-test or one-way ANOVA. The clinicopathologic variables and expressions of *FOXO3* and *IGF1* were analyzed using either the Pearson χ^2^ test (the theoretical frequency was less than five) or the Fisher exact test (the theoretical frequency was less than five). DFS [38] was estimated between the date of surgery and that of cancer recurrence, death, or last follow-up. A censored event included the status of patient who was “alive” without recurrence or death at the last follow-up, whereas failure events were defined as patient death. Kaplan–Meier estimates were performed to illustrate the differences between the survival curves. The same thresholds for *FOXO3* and *IGF1* were utilized for subsequent analysis.

Univariate analysis comparing patient survival between the high and low *FOXO3* and *IGF1* groups was applied while stratifying for individual clinicopathologic variables. Multivariate analysis with the Cox proportional hazards model was performed to estimate the DFS while adjusting for potential confounders. The hazard of individual factors was measured using the hazard rate with 95% confidence interval. All statistical data were analyzed using SPSS (Version 22.0) and GraphPad Prism (GraphPad Software, La Jolla, CA, USA) with *p*-value <0.05.

## 3. Results

### 3.1. Molecular Subtypes Based on Immune Genes and Clinical Characteristics of the Different Subtypes

We assembled a set of immune-associated genes (*n* = 1101) with gene expression profiles. Selected genes from the TCGA cohort were further analyzed in order to distinguish the CESC subtypes. We stratified all samples into k (k = 2–10) differential clusters using the ConsensusClusterPlus (R package). For the result k = 5, the optimal division was achieved, underlying the cumulative distribution function (CDF) curves of the consensus score (Figure 1A). The SigClust analysis elucidated that, among the paired comparisons, the consensus clusters were of significance in k = 5 (Figure 1B). Thus, a set of 303 CESC samples were achieved from the TCGA cohort and stratified into five subtypes (C1, *n* = 85; C2, *n* = 87; C3, *n* = 53; C4, *n* = 51; C5, *n* = 27) based on the whole immune gene expression dataset (Figure 1C). Selected TCGA cases with gene expression data and complete clinical files were analyzed, including age, TNM (tumor, node, metastasis) staging, FIGO stage, neoplasm histologic grade, lymphovascular invasion indicators, and the survival possibility of each subtype. We analyzed the gene expression data and observed the distribution of the clinical features. Using Kruskal–Wallis, the age-group distribution was of significance (*p* = 0.016) (Figure 1D). Among all the subtypes, the average age was the highest in C5 and the lowest in subtype C3. We explored and compared the average age and TNM stage in the molecular subtypes. As compared with the other subtypes, the proportions of T1 in subtype C4, and T3/T4 in subtype C1 were significantly higher, corresponding to the tumor category (*p* < 0.01) (Figure 1E). In addition, the proportions of N0 in subtype C4 and N1 in subtype C1 were significantly higher, corresponding to the node stage (*p* < 0.01) (Figure 1F). Similarly, the proportions of M0 in subtype C1 and M1/M_X_ in subtype C4 were significantly higher, corresponding to metastasis (*p* < 0.01) (Figure 1G). Next, we explored the FIGO stage, the proportions of stage I in sybtype C4 and stage II in subtype C3 were higher, whereas stage III in subtype C4 was significantly lower (*p* < 0.01) (Figure 1H). In addition, the proportions of G1 in subtype C4 and G3 and G4 in subtype C3 were significantly higher (*p* < 0.01) (Figure 1I). The proportions of absence in subtype C4 and presence in subtype C5 were significantly higher (*p* < 0.01) (Figure 1J). Finally, we estimated the survival probability with these five subtypes using the Kaplan–Meier method. The survival probability in sybtype C4 was marginally higher than the other subtypes (Figure 1K, *p* = 0.062). The data indicated that subtype C4 was associated with decreased immune status, whereas subtype C1 was related to an enhanced immune status. Furthermore, the five molecular subtypes from the TCGA cohort showed marginally different prognostic results, with subtype C4 demonstrating the best prognosis.

### 3.2. Estimate Scores Demonstrated a Significant Association with Clinical Outcome in Immune-Related Subtypes

Underlying the ESTIMATE algorithm and immune scores, we calculated the immune, stromal, and ESTIMATE scores among the five CESC subtypes from the TCGA cohort. The immune scores (Figure 2A) and stromal scores (Figure 2B) were both significantly higher in subtype C1 and lower in subtype C4. Moreover, the ESTIMATE scores were also significantly higher in subtype C1 and lower in subtype C4 (Figure 2C). Therefore, the majority of the enhanced immune profiles were included in subtype C1, whereas the majority of the decreased immune profiles were included in subtype C4. In addition, we identified the potential relationship between overall survival (OS) and immune or stromal scores using the ESTIMATE algorithm and Kaplan–Meier method. The set of 303 CESC cases was stratified into a top half and a bottom half (high- vs. low-score groups), corresponding to median overall survival. On the one hand, Kaplan–Meier survival curves showed that cases with either high immune scores (Figure 2D, *p* = 0.021 in the log-rank test) or ESTIMATE scores (Figure 2E, *p* = 0.044) lived longer than those with low scores. On the other hand, cases with high or low stromal scores showed no statistically different median OS (Figure 2F, *p* = 0.14). Thus, subtype C4 demonstrated a downregulation of the immune profile, lower tumor immune/stromal scores, and worse prognosis, whereas subtype C1 showed an upregulation of the immune profile, higher tumor immune/stromal scores, and better prognosis.

### 3.3. Survival Analysis of the CESC Patients in Terms of the Tumor Microenvironment for the TCGA Cohort

Next, we analyzed the prognostic potential and evaluated the distinct outcomes of the CESC patients on the basis of the immune infiltration of B cells, T cells (CD8^+^ and CD4+), macrophages, neutrophils, and dendritic cells (DCs), with or without the presence of a given mutation, via the Kaplan–Meier plotter database (Figure 3). Cases were stratified into a top half and a bottom half (high- vs. low-expression groups) according to the levels of immune infiltration. Kaplan–Meier curves showed that lower levels of immune infiltrates, including CD4^+^ T cells, demonstrated a worse cumulative survival (*p* = 0.027 in the log-rank test). However, there was no correlation between the distinct outcomes for other tumor-infiltrating lymphocytes (TILs).

### 3.4. GO Enrichment Analysis and the Biological Pathways Identified by GSEA for Immune-Related Genes

Next, we performed a functional enrichment analysis to further explore the differentially expressed genes (DEGs). Among a set of 224 immune-related genes, we selected the top 100 genes, based on the absolute value of logFC, that were significantly enhanced in each subtype, and 69 overlapped genes between each subtype were achieved. However, just a few of the overlapped genes could be distinguished in the subtype pairs (Figure 4A). Changes in the biological process (BP) were significantly enriched in chromosome segregation, nuclear division, organelle fission, nuclear chromosome segregation, mitotic nuclear division, and sister chromatid segregation (Figure 4B). Changes in molecular function (MF) were significantly enriched in chromatin binding, protein serine/threonine kinase activity, the catalytic activity of DNA, coupled ATPase activity, and ATPase activity (Figure 4C). Changes in cellular component (CC) were enriched mainly in the chromosomal region, spindle, the centromeric region of the chromosome, condensed chromosomes, and the kinetochore using GO analysis (Figure 4D). The GO analysis suggested that changes in CESC were enriched in nuclear division, chromatin binding, and condensed chromosomes. To explore the HPV-related biological pathways involved in CESC, we analyzed the biological pathways that significantly changed in the samples using a GSEA performed on the samples (GSEA v2.0, http://www.broad.mit.edu/gsea/ accessed on 16 April 2022). The GSEA on gene expression data was primarily related to cellular senescence, the p53 signaling pathway, and viral carcinogenesis which had important correlations with HPV infection (Figure 4E). the GSEA demonstrated that cellular senescence, the p53 signaling pathway, and viral carcinogenesis are critical features of CESC.

### 3.5. Correlation Analysis Performed on the Expression of the Key Genes and Immune Infiltration

Since TILs have an independent influence on sentinel lymph status and prognostic outcome in various types of cancer [39], we assessed the correlations between the immune infiltration level and the gene expression of the key genes, including *PERP*, *BAK1*, *CDK2*, *VDAC1*, *MDM2*, *HDAC1*, *FOXO3*, *AKT3*, and *IGF1*. *PERP*, *BAK1, CDK2*, *VDAC1*, *MDM2*, *HDAC1*, *FOXO3*, *AKT3*, and *IGF* expression were significantly negatively related to tumor purity and positively correlated with T cells (CD^8^ and CD^4)^, macrophages, neutrophils, and DCs in CESC, but not significantly correlated with B cells (Figure 5A–I).

### 3.6. Survival Analysis of the Expressions of the Key Genes in the Tumor Microenvironment from the TCGA Cohort

To explore the clinical relevance of the key genes in the tumor immune subsets of CESC cases, we generated a Kaplan–Meier curve to compare the expressions of *PERP* (Figure 6A), *BAK1* (Figure 6B), *CDK2* (Figure 6C), *VDAC1* (Figure 6D), *MDM2* (Figure 6E), *HDAC1* (Figure 6F), *FOXO3* (Figure 6G), *AKT3* (Figure 6H), and *IGF1* (Figure 6I) at different expression levels of immune infiltration. High expression of *VDAC1* (Figure 6D) and *FOXO3* (Figure 6G) suggested a significantly poor outcome in the log-rank test (*p* = 0.045 and *p* = 0.048, respectively). However, low expression of *IGF1* (Figure 6I) also showed a poor outcome (*p* = 0.047). In addition, high expression of *PERP* (Figure 6A), *BAK1* (Figure 6B), *CDK2* (Figure 6C), *MDM2* (Figure 6E), *HDAC1* (Figure 6F), and *AKT3* (Figure 6H) predicted poor overall survival, but it was not of statistical significance (*p* > 0.05). *VDAC1*, *FOXO3,* and *IGF1* might be the crucial differential expression genes in CESC.

### 3.7. Correlation between Gene Copy Number Variation and Immune Infiltration Abundance in CESC

To investigate whether the mutation of the key genes, including *PERP* (Figure 7A), *BAK1* (Figure 7B), *CDK2* (Figure 7C), *VDAC1* (Figure 7D), *MDM2* (Figure 7E), *HDAC1* (Figure 7F), *FOXO3* (Figure 7G), *AKT3* (Figure 7H), and *IGF1* (Figure 7I) were involved in the immune infiltration portion of the infiltrates, we divided the key genes into six groups underlying different copy number variation (CNV) conditions to explore the different infiltration levels of six immune cells between six groups. Mutation in the key genes was negatively related to the infiltration level of DCs, and most of the other infiltrates, including T cells (CD8^+^ and CD4^+^), macrophages, and neutrophils, decreased significantly. *p*-value significance codes: *p* < 0.1, * 0.01 ≤ *p* < 0.05, ** 0.001 ≤ *p* < 0.01, **** 0 ≤ *p* < 0.001.

### 3.8. Correlations between the Key Genes and Prognosis in the East Hospital Cohort

We performed and verified the key genes of patients for further validation in the East Hospital (EH) cohort using tissue chip. Six key genes were located within malignant tumor cells and enriched predominantly in the cytoplasm of tumor cells, including *PERP*, *BAK1*, *VDAC1*, *FOXO3*, *AKT3*, and *IGF1* (Figure 8A). Punctate staining was also identified in the membrane nuclei of some tumor cells. However, *MDM2*, *HDAC1*, and *CDK2* were not expressed in tissue chip. Significant differences were identified for age (11.989 versus 13.123, *p* < 0.0001), T category (HR 14.746 versus 12.926, *p* < 0.0001), N stage (HR 15.670 versus 11.657, *p* < 0.0001), and recidivation (HR 52.145 versus 60.813, *p* < 0.0001), in light of the distinct *FOXO3* and *IGF1* expression levels. However, *FOXO3* and *IGF1* gene expressions were not statistically different in terms of pathology grade (HR 0.930 versus 0.894, *p* = 0.335 versus *p* = 0.344) or HPV infection (HR 0.393 versus 0.673, *p* = 0.531 versus *p* = 0.412). In addition, there was no significant difference between *FOXO3* and *IGF1* expression levels (HR 1.138 versus 1.700, *p* = 0.286 versus *p* = 0.192). We performed Kaplan–Meier survival curves to further analyze the associations between the disease-free survival (DFS) of patients in the East Hospital (EH) cohort and key genes, including *PERP* (Figure 8B), *BAK1* (Figure 8C), *VDAC1* (Figure 8D), *FOXO3* (Figure 8E), *AKT3* (Figure 8F), and *IGF1* (Figure 8G) in CESC. Low expression of *FOXO3* (*p* = 0.0423, Figure 8E) and high expression of *IGF1* (*p* = 0.0438, Figure 8G) demonstrated significantly poor DFS in the log-rank test. In the univariate analysis, age (hazard rate (HR) 6.101, *p* < 0.001), T category (HR 12.73, *p* < 0.001), N stage (HR 4.658, *p* < 0.001), and *FOXO3* (HR 2.473, *p* = 0.0423), and *IGF1* expression (HR 0.454, *p* = 0.0438) were significantly correlated with DFS, whereas pathology grade (HR 1.149, *p* = 0.734), HPV infection (HR 1.016, *p* = 0.976), and *BAK1* (HR 0.872, *p* = 0.6924), *VDAC1* (HR 1.377, *p* = 0.3943), *AKT3* (HR 1.435, *p* = 0.3282), and *PERP* expression (HR 0.855, 95%, *p* = 0.6892) were not related to DFS. In the multivariate analysis, only age (HR 7.959, *p* = 0.007), N stage (HR 6.892, *p* = 0.013), and *FOXO3* expression (HR 11.611, *p* = 0.047) were independent prognostic factors for CESC (Table 1 and Table 2).

## 4. Discussion

CESC generally presents with HPV infection and is the second most common gynecological malignancy. However, its pathological mechanism remains unclear. Although advances in treatment strategies have improved overall patient prognosis, metastasis and recurrence still pose challenges in clinical setting. Therefore, understanding the molecular mechanisms of the progression of CESC, including metastasis, would aid in the development of effective diagnostic and targeted therapies. A number of studies have reported that the intricate microenvironment sustained by HPV infection increases the risk for CESC progression and participates in a variety of signaling pathways, including those related to cell adhesion and EMT [40]. In addition, partly based on the interaction between cancer cells and the tumor microenvironment [41], cancer is best defined as an ecosystem in which tumor cells interact with specific environments such as immune cells and interstitial cells, adapt to each other, and even coevolve in a multidimensional space-time manner [42]. Here, we present a case study of CESC patients for systematically analyzing coexisting CESC molecular subtypes and the clinical significance underlying global immune genes with downloaded data from TCGA. We used gene expression data to validate that the five molecular subtypes potentially underlie the pathological process of CESC. To further evaluate the immunological relationships between the CESC samples and the molecular subtypes, we determined the clinical characteristics, immune infiltration levels, and survival outcomes. In addition, we selected all DEGs and performed a functional enrichment analysis. Furthermore, correlations between key genes with immune infiltrates, gene copy number variation, and outcome were validated.

Numerous studies have revealed several molecular subtypes of CESC according to genome-wide profiles [43]. In this study, we explored global gene expression data as part of a more comprehensive study of the immune landscape of CESC. Subtype C4 was associated with decreased immune status, whereas subtype C1 was related to an enhanced immune status, which exhibited a positive relationship with immune-related cell expression features. Furthermore, the five molecular subtypes from TCGA cohort showed marginally different prognostic results, with subtype C4 demonstrating the best prognosis. Therefore, we hypothesized that the immune-enhanced and immune-reduced subtypes coexist in the tumor microenvironment of CESC. These subtypes showed remarkably different expression data related to mutagens, immune component scores, and clinical prognosis. We further investigated the prognostic potential of the immune infiltration levels and the relationship with clinical prognosis in the five subtypes in the microenvironment of CESC. The clinical outcomes of the CESC patients were closely related with the immune infiltrating levels of CD4^+^ T cells. As expected, among all the subtypes, immune scores, stromal scores, and ESTIMATE scores were highest in subtype C1 and lowest in subtype C4.

To identify the most effective diagnostic biomarkers, we performed a synthetic analysis and 224 DEGs were identified. To analyze the function enrichment of DEGs, we annotated CESC with GO terms and performed GSEA as represented by DEGs. A set of biological processes and pathways, including the mitotic cycle, chromosomes, and the catalytic activity of nuclear DNA, were suggested to be involved in CESC. As such, the DEGs may locate in the nucleus and participate in cell cycle processes in order to promote cellular proliferation by enhancing DNA catalytic activity in CESC. Biological processes, the mitotic cycle, chromosome segregation, nuclear division, organelle fission, nuclear chromosome segregation, mitotic nuclear division, and sister chromatid segregation were the most significant. Gene expression analysis, GO function enrichment, and pathway enrichment analysis indicated that the occurrence and progression of CESC are considerably complex in terms of gene expression, multi-cellular processes, and coexisting immune microenvironments. Factors in the immune microenvironment of CESC have been previously reported [40], and our results were consistent with previous findings.

Previous studies have demonstrated that CESC presents with a positive regulation of transcriptional, DNA template, nucleoplasm, and intrinsic apoptotic signaling pathways [44]. The DEGs identified in the immune microenvironment, including *IL1R2* [45], *CDK1* [46], *CXCL14* [47], and *RANKL* [48] have been further revealed as functional oncogenes involved in the cell cycle of CESC by PCR and Western blot analysis. Furthermore, we performed GSEA and investigated the biological functions of the DEGs in CESC. In CESC, dysregulation of signaling pathways involved in oocyte meiosis, cell cycle, p53 signaling, and progesterone-mediated oocyte maturation have been reported [46]. In addition, we identified cellular senescence, p53 signaling, and viral carcinogenesis to be implicated in CESC. It is well known that about 90% of CESC cases are caused by HPV infection, which plays an important role in the development of carcinogenesis [49]. After infection, disrupted p53-mediated regulation of the cell cycle and apoptosis has been shown to be inhibited by E6 and E7 viral oncogenic proteins [50]. E6/E7 oncogene is associated with the rapid regeneration of p53 [51] and pRb anti-proliferative protein [50], phenotypically resulting in cellular senescence [50]. The tumor suppressor p53 is a key gene with regard to cellular proliferation and apoptosis and exhibits a relatively high prognostic value in CESC. Some persistent viral oncogenes can inactivate p53 and pRb, leading to increased genomic instability, accumulation of somatic mutations and, in some cases, integration of HPV into the host genome [7]. In brief, the findings of our study suggested that HPV oncogenes induced mitotic processes and prevented the progression of the cell cycle through the p53 pathway.

In addition, we performed an overall survival analysis of patients on the basis of the immune infiltration levels of infiltrates with or without the presence of a given mutation in CESC. Indeed, accumulating evidence has suggested that reversed CD4/CD8 ratios are closely correlated with clinical outcome in patients with CESC [52,53]. In recent studies, it has been observed that DEG expression was closely correlated with diverse immune infiltration levels in CESC. Moreover, there appears to be a strong positive relationship between the prognosis of key genes and immune infiltration levels of CD4^+^ T cells. TILs have been regarded as an independent factor of lymph status and prognosis in tumors. In this study, we investigated, in detail, the correlations between immune infiltration levels of key genes and clinical outcomes. We performed and verified the key genes of patients for further validation in the East Hospital (EH) cohort using tissue chip. Six key genes were located within malignant tumor cells and enriched predominantly in the cytoplasm of tumor cells, including *PERP*, *BAK1*, *VDAC1*, *FOXO3*, *AKT3*, and *IGF1.* Significant differences were identified for age, T category, N stage, and recidivation, in light of the distinct *FOXO3* and *IGF1* expression levels. However, *FOXO3* and *IGF1* gene expressions were not statistically different in terms of pathology grade or HPV infection. In addition, we performed Kaplan–Meier survival curves to further analyze the associations between the disease-free survival (DFS) of patients in the East Hospital (EH) cohort and key genes. We demonstrated high protein expression of *FOXO3* and low expression of *IGF1* suggested poor clinical outcome. Meanwhile, mutation of the key genes was negatively related to the infiltration levels of most of the other infiltrates. CD4^+^ T cells are crucial in immune response protection, and the memory subtype endows the host with increased secondary immune reactions [54,55]. HPV-related lesions have been shown to be removed by the anti-infective immune responses of T cells [56]. During the clearance of HPV, T cells (CD4^+^ and CD8^+^) appear to be the major anti-infective cells [57]. This conclusion has been well exemplified by other findings that have demonstrated that the density and distribution of immune T cells depend on the malignant potential of HPV-related lesions, with increases in circulating CD4^+^ T cell populations associated with the progression of CESC [58,59]. Therefore, CD4^+^ T cells may be essential regulators for controlling HPV-related cervical lesions and preventing carcinogenesis.

Accumulating evidence has shown that *FOXO* increases antitumor activity by negatively regulating immunosuppressive protein expression, including that of PD-L1 and vascular endothelial growth factor (VEGF) [60,61]. Thus, *FOXO3* has been proposed to be a regulator that is intrinsically involved in tumor immunity, homeostasis, and immunocytes growth, including T cells, NK cells, and DCs [62,63]. A recent study has indicated that HPV protein blocks the TGFβ/miR-182/BRCA1 and *FOXO3* path in head and neck squamous cell carcinoma cell [64]. Moreover, *FOXO3*, which is considered to be a regulator for *FOXM1*, has been shown to participate in the development of cervical cancer and the lactate-rich microenvironment during HPV infection in cervical squamous carcinoma cell [40]. However, the role of *FOXO3* in CESC and HPV infection in the immune microenvironment remains to be elucidated.

The *IGF* family plays a significant role in cell differentiation, proliferation, carcinogenesis, and apoptosis [65,66], and it has become an attractive therapeutic target [67]. Preclinical studies have reported that the IGF1/IGF1 receptor axis participates in multiple cross-talk signaling pathways involved in the maintenance of keratinocytes [68] as a result of HPV infection. *IGF1* has also been elucidated as a dictator CD4^+^ T cell by enhancing ERβ transcriptional activity [69] and an immuno editor that eradicates new emerging tumor cells in the immune system [70]. Accordingly, an important approach in cancer therapy participates in the inhibition of the *IGF1* pathway [71]. In addition, it has been suggested that, in CESC, *IGF1* facilitates the stabilization of direct or indirect interactions with E6 and E7 HPV proteins [72]. Other evidence has shown that Bak-1 is involved in the DHA-induced autophagy observed in HeLa cells [73] and the ERK-mediated CDK2/cyclin-A signaling pathway, and induces apoptosis and G1/S arrest in Hela and Caski cells [74]. Furthermore, *VDAC1* has been shown to prevent the progression of HPV-induced cervical cancer [75] and DHX9–lncRNA–MDM2 interactions regulate tumor cell invasion and the angiogenesis of CESC [22]. *HDAC1* increases during immune-editing and contributes to immune refractory cancers, including CESC. Silencing *AKT3* in Hela/DDP cells may enhance their sensitivity to cisplatin [76]. Moreover, *PERP* inhibition has been associated with apoptosis and VEGF suppression in lung cancer [77]. Therefore, these key genes in the immune microenvironment may be potential prognostic factors and biomarkers for predicting the occurrence and metastasis of CESC.

This study had several limitations, including the relatively low number of clinical characteristics available, including relatives and the pathological features of patients with CESC in the genetic subgroup study. Less data may also have contributed to unilateral results and high false positives. Future research should include a larger sample size and further validation of our findings to evaluate whether the deviation is caused by the difference in the sequencing race at home and abroad or the insufficient sample size.

## 5. Conclusions

In conclusion, our findings indicated that immune-related genes in the tumor microenvironment can be stratified into five subtypes, with potential mechanisms of immunological evasion in CESC that can distinguish immunophenotype features, immune checkpoint molecules, and clinical prognosis indicators, including immune and stromal scores. In addition, specific functional pathways, including cellular senescence, p53 signaling, and viral carcinogenesis may be closely associated with HPV infection and the coexisting microenvironment subtypes. These findings may aid in promoting the improvement of CESC immunotherapy strategies.

## Figures and Tables

**Figure 1 cancers-15-01419-f001:**
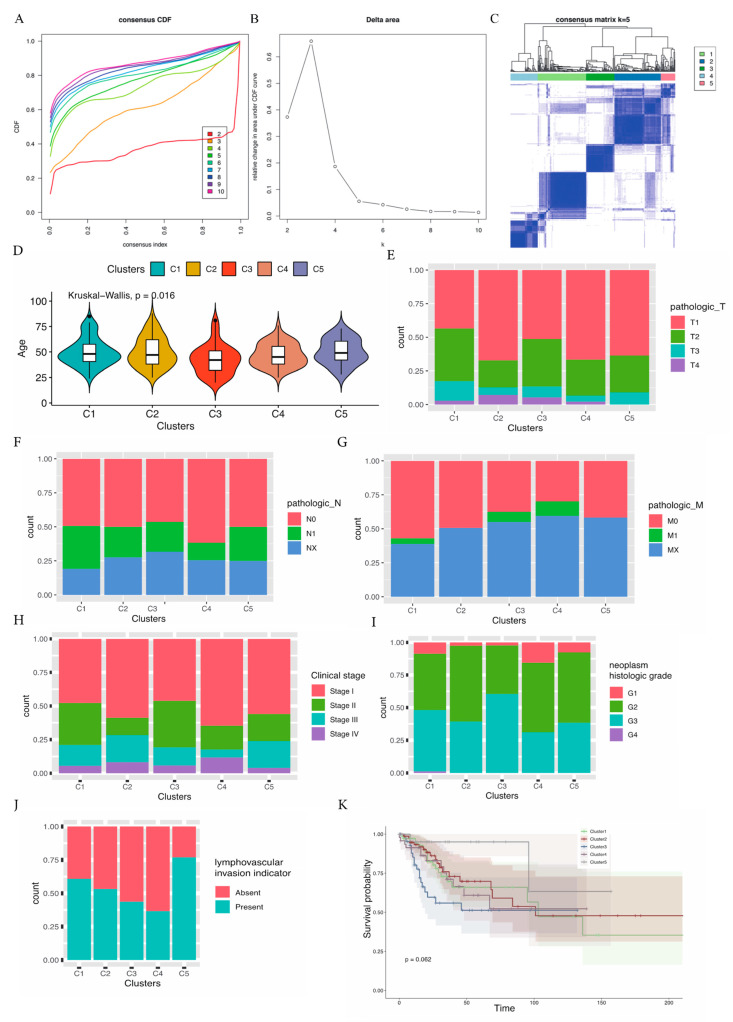
Identification of immune cluster related to molecular subtypes of CESCs in the TCGA cohort, the expression profile analysis of five CESC subtypes and factor analysis of five subtypes underlying clinical pathological characteristics; (**A**) K generated from the number of clusters; (**B**) the CDF Delta area curve of all samples when k = 5; (**C**) the consensus score matrix of all CESC samples when k = 5. Relationship between five subtypes and age (**D**), T stage ratio (**E**), regional lymph node ratio (**F**), metastasis ratio (**G**), FIGO Stage ratio (**H**), neoplasm histological grade (**I**), lymphovascular indicator invasion (**J**), and overall survival analysis (**K**), in the five CESC subtypes.

**Figure 2 cancers-15-01419-f002:**
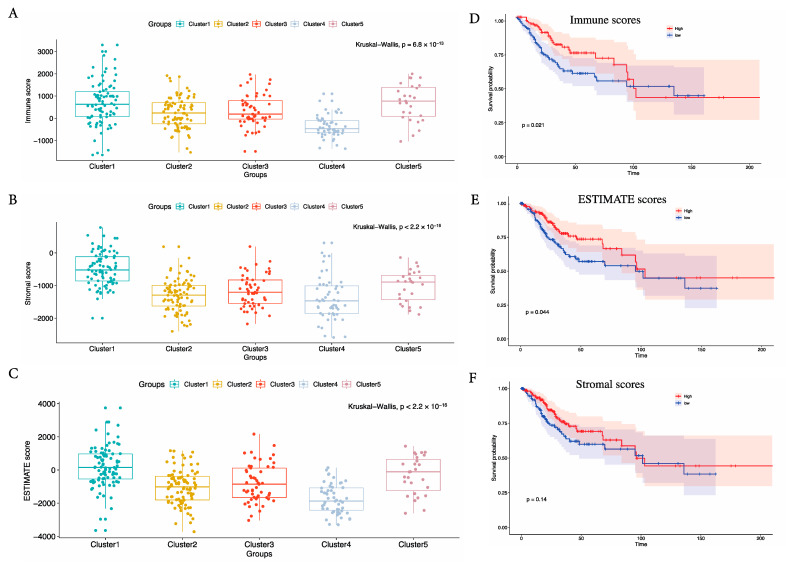
The immune infiltration levels in the five CESC subtypes from the TCGA cohort and survival analysis for the five subtypes of CESCs. Kaplan–Meier curves showed the distinct outcomes of patients underlying immune scores (**A**), stromal scores (**B**), and ESTIMATE scores (**C**). *p* < 0.05 was considered to be statistically significant. The immune infiltration levels in the five CESC subtypes from the TCGA cohort underlying the immune scores (**D**), the estimate scores (**E**), and the stromal scores (**F**). *p* < 0.05 was considered to be statistically significant.

**Figure 3 cancers-15-01419-f003:**
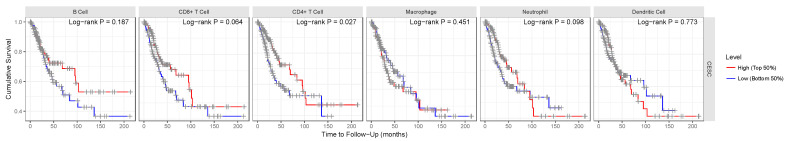
Survival analysis of the CESC patients in the tumor microenvironment for the TCGA cohort. Kaplan–Meier curves showed the distinct outcomes of the CESC patients based on infiltrating levels of B cells, CD8^+^ T cells, CD4^+^ T cells, macrophages, neutrophils, and dendritic cells in CESC. *p* < 0.05 was considered to be statistically significant.

**Figure 4 cancers-15-01419-f004:**
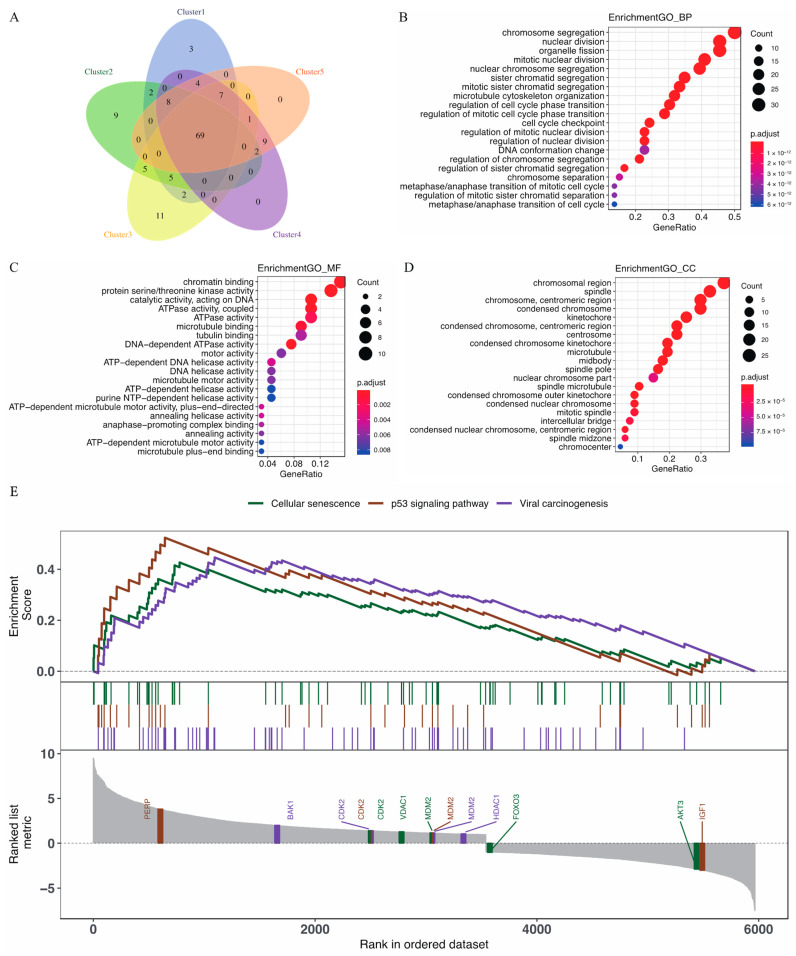
Functional enrichment analysis and gene set enrichment analysis (GSEA) in the CESC cohort. (**A**) Venn diagram of the overlapping genes across the five subtypes; (**B**–**D**) bubble plots demonstrating distribution of the gene ontology annotations for CESC in BP (**B**), MF (**C**), and CC (**D**) (BP, biological process; MF, molecular function; CCs, cellular component); (**E**) GSEA shows that cellular senescence, the p53 signaling pathway, and viral carcinogenesis are enriched in CESC.

**Figure 5 cancers-15-01419-f005:**
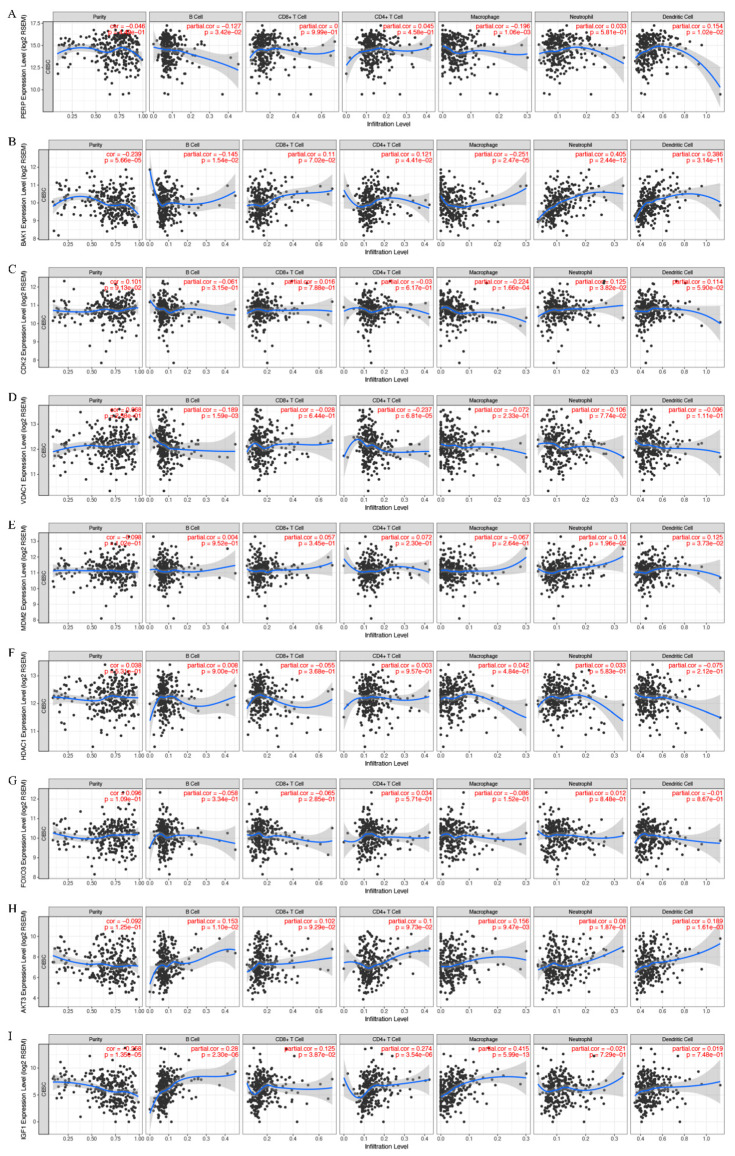
Correlations between the immune infiltration level and the gene expression of the key genes in CESC: (**A**) PERP; (**B**) BAK; (**C**),CDK2; (**D**) VDAC1; (**E**) MDM2; (**F**) HDAC1; (**G**) FOXO3; (**H**) AKT3; (**I**) IGF.

**Figure 6 cancers-15-01419-f006:**
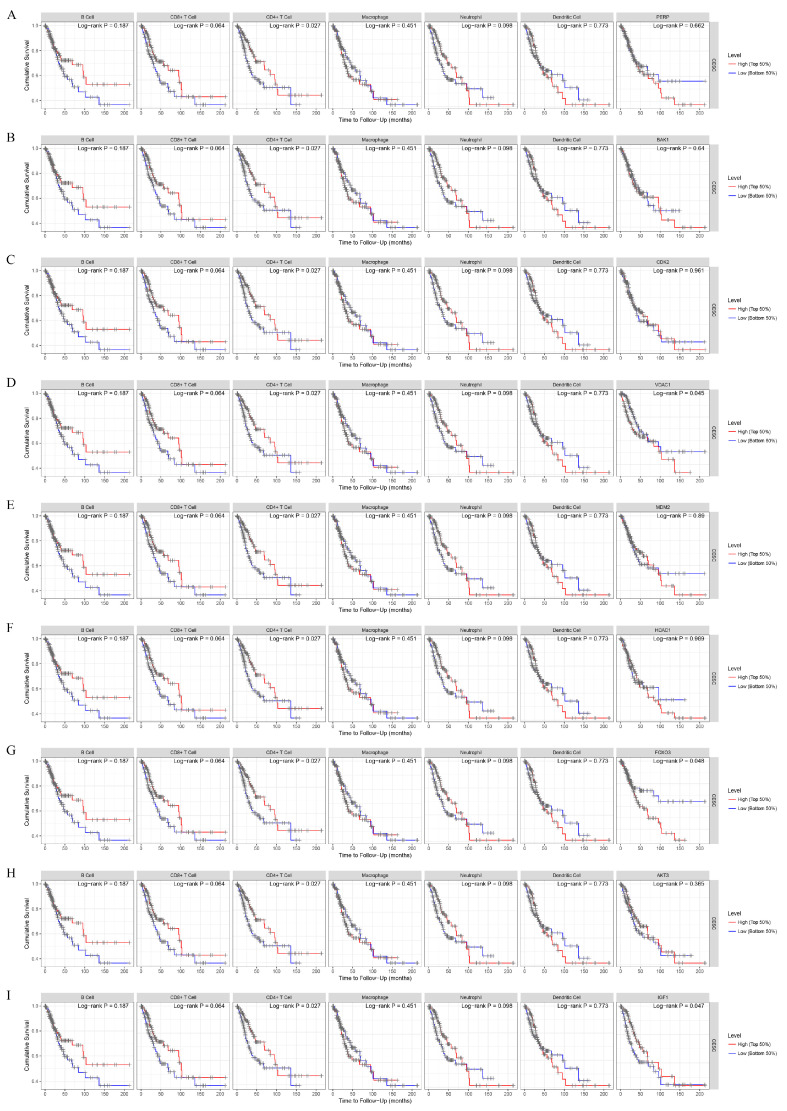
Survival analysis of key gene expressions in the tumor microenvironment of CESC in the TCGA cohort. Kaplan–Meier survival curves comparing the high and low expressions of PERP (**A**), BAK1 (**B**), CDK2 (**C**), VDAC1 (**D**), MDM2 (**E**), HDAC1 (**F**), FOXO3 (**G**), AKT3 (**H**) and IGF1 (**I**) in CESC from the TCGA cohort.

**Figure 7 cancers-15-01419-f007:**
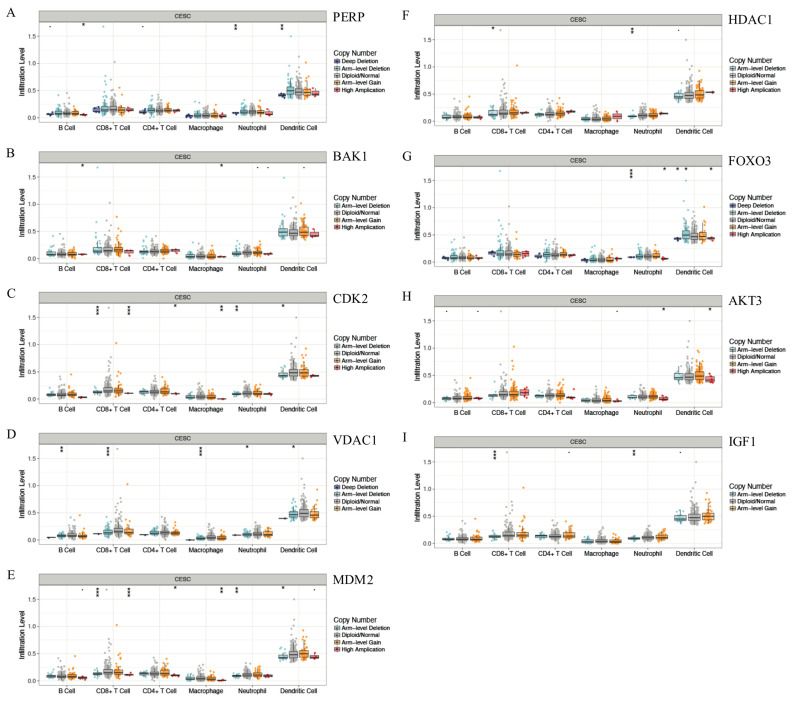
Correlations of the immune infiltration abundance and the copy number variation of the key genes, including PERP (**A**), BAK1 (**B**), CDK2 (**C**), VDAC1 (**D**), MDM2 (**E**), HDAC1 (**F**), FOXO3 (**G**) and AKT3 (**H**), and IGF1 (**I**), according to different CNV conditions to explore the different infiltration levels of six immune cells among groups in CESC.

**Figure 8 cancers-15-01419-f008:**
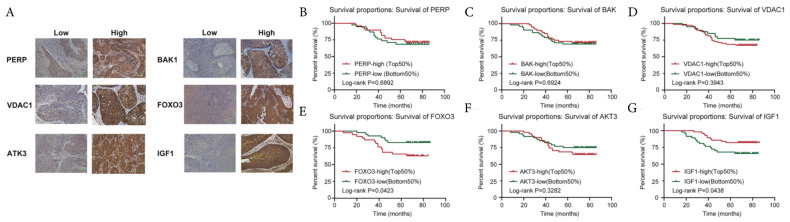
Survival analysis of the expression of key genes in the East Hospital (EH) cohort: (**A**) Immunohistochemical characterization of key gene expressions of CESC specimens in the EH cohort; (**B**–**G**) Kaplan–Meier survival curves of CESC patients for DFS as compared with the high and low expression of PERP (**B**), BAK1 (**C**), VDAC1 (**D**), FOXO3 (**E**), AKT3 (**F**) and IGF1 (**G**), in the EH cohort. Original magnifications ×100 (lower panels). EH cohort, East Hospital cohort.

**Table 1 cancers-15-01419-t001:** Associations among FOXO3, IGF1 expression, and clinicopathologic features in the EH cohort.

Variable	No. (%)	FOXO3 Expression		χ^2^	Log-Rank-*p* No. (%)	No. (%)	IGF1 Expression		χ^2^	Log-Rank-*p* No. (%)
		Low (%)	High (%)				Low (%)	High (%)		
Age				11.989	<0.001				13.123	<0.001
<50	43 (75.44)	22 (75.86)	21 (75.00)			43 (75.44)	20 (76.92)	23 (74.19)		
≥50	14 (24.56)	7 (24.14)	7 (25.00)			14 (24.56)	6 (23.08)	8 (25.81)		
T category				14.746	<0.001				12.926	0.002
1	31 (54.39)	17 (58.62)	14 (50.00)			31 (54.39)	12 (46.15)	19 (61.29)		
2	14 (24.56)	6 (20.69)	8 (28.57)			14 (24.56)	5 (19.23)	9 (29.03)		
3/4	12 (21.05)	6 (20.69)	6 (21.43)			12 (21.05)	9 (34.62)	3 (9.68)		
N stage				15.670	<0.001				11.657	<0.001
0	46 (80.70)	23 (79.31)	23 (82.14)			46 (80.70)	17 (65.38)	29 (93.55)		
1	11 (19.30)	6 (20.69)	5 (17.86)			11 (19.30)	9 (34.62)	2 (6.45)		
Pathology grade				0.930	0.335				0.894	0.344
I–II	16 (23.53)	8 (27.59)	6 (21.43)			14 (24.56)	5 (19.23)	9 (26.03)		
III	52 (76.47)	21 (72.41)	22 (78.57)			43 (75.44)	21 (80.77)	22 (70.97)		
HPV infection				0.393	0.531				0.673	0.412
Negative	10 (17.54)	6 (20.69)	4 (14.29)			10 (17.54)	6 (23.08)	4 (12.90)		
Positive	47 (82.46)	23 (79.31)	24 (85.71)			47 (82.46)	20 (76.92)	27 (87.10)		
Recidivation				52.145	<0.001				60.813	<0.001
Negative	44 (77.19)	25 (86.21)	19 (67.86)			44 (77.19)	20 (76.92)	24 (77.42)		
Positive	13 (22.81)	4 (13.79)	9 (32.14)			13 (22.81)	6 (23.08)	7 (22.58)		
IGF1 expression				1.138	0.286				1.700	0.192
Low	26 (45.61)	17 (58.62)	9 (32.14)			29 (50.88)	17 (65.38)	12 (38.71)		
High	31 (54.39)	12 (41.38)	19 (67.86)			28 (49.12)	9 (34.62)	19 (61.29)		

**Table 2 cancers-15-01419-t002:** Associations with DFS and clinicopathologic characteristics in the EH cohort using Cox regression.

Factor	Univariate Analysis		Multivariate Analysis	
	HR (95% CI)	*p*	HR (95% CI)	*p*
Age	6.101 (2.724, 13.67)	<0.001	7.959 (1.742, 36.364)	0.007
T category	12.73 (6.073, 26.67)	<0.001	16.414 (0.844, 319.14)	0.065
N stage	4.658 (1.659, 13.08)	<0.001	6.892 (1.512, 31.408)	0.013
Pathology grade	1.149 (0.5268, 2.504)	0.7346	0.296 (0.057, 1.523)	0.145
HPV infection	1.016 (0.3537, 2.921)	0.9759	5.447 (0.430, 68.967)	0.191
IGF1	0.454 (0.208, 0.989)	0.0438	0.090 (0.008, 1.081)	0.058
FOXO3	2.473 (1.046, 5.845)	0.0423	11.611 (1.033, 130.506)	0.047
BAK	0.872 (0.439, 1.731)	0.6924	0.817 (0.081, 8.286)	0.864
VDAC1	1.377 (0.680, 2.787)	0.3943	0.439 (0.069, 2.784)	0.382
AKT3	1.435 (0.701, 2.936)	0.3282	12.481 (0.214, 729.339)	0.357
PERP	0.855 (0.393, 1.862)	0.6892	0.515 (0.143, 2.650)	0.515

## Data Availability

To analyze the roles of DEGs, RNA-seq data and relevant clinical data were downloaded from the Timer database (https://cistrome.shinyapps.io/timer/ accessed on 1 March 2022) and TCGA (https://tcga-data.nci.nih.gov/tcga/ accessed on 9 January 2022). We analyzed the biological pathways that significantly changed in the samples using GSEA (GSEA v2.0, http://www.broad.mit.edu/gsea/ accessed on 16 April 2022). All data are available to be released. The datasets used and/or analyzed during the current study are available from the corresponding author on reasonable request.

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
