# Peer review of "Prognostic Value and Immune Infiltration of HPV-Related Genes in the Immune Microenvironment of Cervical Squamous Cell Carcinoma and Endocervical Adenocarcinoma"

_cancers, 2023, doi:10.3390/cancers15051419_

Round 1
Reviewer 1 Report
The authors demonstrated that these findings provided novel insight into the relationship between the immune microenvironment and CESC. A well-written manuscript. However, several points should be noted as below.
1)How about HPV status in TCGA database and 115 Chinese patients diagnosed with CESC? If HPV16 +, HPV18 + and HPV-negative were present in TCGA database, for example, differences between the immune landscape need to be further examined.
2)The authors demonstrated that their findings “may provide guidance for developing potential immune-therapeutic targets and biomarkers for CESC” (Abstract). The differentially expressed genes (DEG), immune infiltration, and drug sensitivity need to be analyzed.
3)Table 1 is missing?
4)The images in “Figure 11. Immunohistochemical..” were very obscure. The immunohistochemical images of serial sections in a a tumor are suggested to be provided. In addition, correlation between the immune infiltration level and key genes expression also need to be detected (with IHC images).
5)In fact, partly based on the interaction of cancer cells and tumor microenvironment, cancer may best be conceptualized as an ecosystem (â‘ Kenny PA, Nelson CM, Bissell MJ. The Ecology of Tumors: By perturbing the microenvironment, wounds and infection may be key to tumor development. Scientist. 2006;20(4):30.). An interesting paper recently proposed that the initiation and progression of human diseases including cancer could be a spatiotemporal ecological process, in such complex systems, parenchymal cells interplay with their specific environments include immune cells, interstitial cells, adapt to each other and even coevolve in a multidimensional spatiotemporal manner (â‘¡ Nasopharyngeal Carcinoma Ecology Theory: Cancer as Multidimensional Spatiotemporal “Unity of Ecology and Evolution” Pathological Ecosystem. Preprints. 2022; 2022100226. Please check, https://www.preprints.org/manuscript/202210.0226/v3) These views should be better updated.
Author Response
Response:
- Thank you for your comments and suggestion. The clinical information in TCGA and Chinese samples does not have the detailed information source of this piece, so although we are very interested, we cannot further study this piece of data at present. We will supplement the information collection of this part in the follow-up clinical work, so as to look forward to further study in the follow-up study.
-
Thank you for your comments and suggestion. We analyzed this part in the discussion section (please see line 428-442, text highlighted in gray).
-
I'm sorry for this silly mistake, which caused unnecessary trouble when reading the article. We have made corresponding corrections in the manuscript.
-
I'm sorry for this silly mistake, which caused unnecessary trouble when reading the article. We have reuploaded images of higher quality.
-
Thank you for your comments and suggestion. We have made corresponding corrections in the manuscript (please see line 415-419, text highlighted in gray).
Reviewer 2 Report
Comments to authors:
In this manuscript, with comprehensive bioinformatic analysis of the data from TCGA and East Hospital cohort, Gan et al. identified the prognostic value of immune-related genes on CESC, which provide novel insights into the relationship between the immune microenvironment and CESC. Rational methods were applied, and the findings showed certain novelty and significance. However, there are some flaws and issues that need further clarification.
Major points:
1. Please include a flowchart in the text body or in the supplementary materials.
2. Please present the results in a more logical way, which will make the manuscript easier to understand. Based on the major findings, some of the figures could be combined to make it clear. For example, Figure 1 and 2 could be combined into one, so is Figure 3 and 5, Figure 6 and 7, Figure 4 and 8, Figure 9 and 10, Figure 11 and 12.
3. The authors investigated the immune microenvironment and CESC, but not HPV-related genes. All the results about HPV is in the pathway analysis. Maybe, an alternative title will fit the content better
4. Please provide the missing table 1 and table 2 in the supplementary material. The results of univariate and multivariate cox regression analysis should also be included in the supplementary file.
Minor points:
1. Line 288: How did 244 immune-related genes identified from 1101 genes? Please describe it in detail.
2. Line 289: Were the “top 100 genes” sorted by P value or fold change? Please describe it clearly.
3. How many DEGs is the Go enrichment analysis performed on?
4. Line 307-308: “we analyzed the biological pathways that significantly changed in the samples using GSEA”. Is the GSEA performed on the DEGs or the samples? Please point it out clearly
5. Line 320: How did the authors get the “key genes”? Are they from previous published works for certain analysis?
6. Line 353: please provide the full name of the abbreviation CNV.
7. Result 3.11 and 3.12: It showed the validation of 6 out of the 9 key genes in the clinical cohort. What about the other 3?
8. Figure 11: There are no markers of A-F on the pictures of figure 11.
9. Line 368: The IHC result for IGF1 were not mentioned.
10. The “Discussion” need some improvement.
In the survival analysis of the key genes in TCGA cohort, the authors found that “High expression of VDAC1) and FOXO3 suggested a greatly poor outcome in log-rank test. Low expression of IGF1 also showed a poor outcome” (showed in figure 9). However, controversial results were revealed in the East Hospital cohort (showed in figure 12). It would be necessary and interesting to have some discussion about this.
The significance of the validation on the East Hospital cohort would need further discussion.
Author Response
Response:
Major points
- Thank you for your comments. We made a flow chart as a supplementary Figure to upload.
-
Thank you for your comments and suggestion. We have made corresponding corrections in the manuscript. We integrated the contents of Figure 1 and 2, Figure 3 and 5, Figure 6 and 7, Figure 11 and Figure 12, and the corresponding figures were also integrated, but in consideration of the logic and consistency of the article, we did not integrate Figure 4 , Figure 8, Figure9, Figure10.
-
I'm sorry for this silly mistake, which caused unnecessary trouble when reading the article. In fact, the study screened three HPV infection-related pathways through GSEA analysis, and the key genes are enriched in the HPV-related pathways, and the subsequent analysis focused on the relationship between these genes and immunity. So we believe that HPV should exist in the title.
-
I'm sorry for this silly mistake, which caused unnecessary trouble when reading the article. We have made corresponding corrections in the manuscript.
Minor points
- I'm sorry for this silly mistake, which caused unnecessary trouble when reading the article. We have made corresponding corrections in the manuscript (please see line 443-444, text highlighted in gray).
-
I'm sorry for this silly mistake, which caused unnecessary trouble when reading the article. We have made corresponding corrections in the manuscript (please see line 297, text highlighted in gray).
-
Thank you for your comments. We are sorry for the confusion caused by these errors. We have made an explaination this in the manuscript (please see line 296-298, text highlighted in gray).
-
I'm sorry for this silly mistake, which caused unnecessary trouble when reading the article. We have made corresponding corrections in the manuscript (please see line 310-311, text highlighted in gray).
-
I'm sorry for this silly mistake, which caused unnecessary trouble when reading the article. We have made corresponding corrections in the manuscript (please see line 313, text highlighted in gray). The study screened three HPV infection-related pathways through GSEA analysis, and the key genes are enriched in the HPV-related pathways. Meanwhile, we analyzed the research status of each gene one by one in the discussion section(please see line 507-534, text highlighted in gray).
-
I'm sorry for this silly mistake, which caused unnecessary trouble when reading the article. We have made corresponding corrections in the manuscript (please see line 358, text highlighted in gray).
-
Thank you for your comments. We are sorry for the confusion caused by these errors. Since the other three genes were not expressed in immunohistochemistry, subsequent analysis was not performed. We have made corresponding corrections in the manuscript (please see line 373-374, text highlighted in gray).
-
Thank you for your comments. We are sorry for the confusion caused by these errors. We have made corresponding corrections in the Figures. We merged Figure 11 and Figure 12 into Figure 8, and Figure 11 became Figure 8A.
-
I'm sorry for this silly mistake, which caused unnecessary trouble when reading the article. We have made corresponding corrections in the manuscript (please see line 372, text highlighted in gray).
- Thank you for your comments. We are sorry for the confusion caused by these errors. We have made corresponding corrections in the manuscript (please see line 542-543, text highlighted in gray). We have added the shortcomings of this study to the tailender paragraph of the discussion. The significance of verification is also discussed in the discussion section(please see line 486-495, text highlighted in gray).
Round 2
Reviewer 1 Report
No other question
Reviewer 2 Report
The manuscript has been sufficiently improved to be published in Cancers.